# Associations between Age at Menarche and Dietary Patterns with Blood Pressure in Southwestern Chinese Adults

**DOI:** 10.3390/nu14081610

**Published:** 2022-04-12

**Authors:** Ting Chen, Deqiang Mao, Liling Chen, Wenge Tang, Xianbin Ding

**Affiliations:** 1Department of Non-Communicable Disease Control and Prevention, Chongqing Center for Disease Control and Prevention, Chongqing 400042, China; tingchencqjk@126.com (T.C.); maodq@yeah.net (D.M.); mbcllgz@163.com (L.C.); 2School of Public Health and Management, Chongqing Medical University, Chongqing 400016, China

**Keywords:** age at menarche, dietary pattern, hypertension

## Abstract

The aim of our study was to examine the relationship between age at menarche (AM) and hypertension and to evaluate whether different dietary patterns have an effect on associations between AM and hypertension in a large-scale Han Chinese population in southwest China. A cross-sectional study was performed that included 44,900 participants from 17 districts in southwest China from September 2018 to January 2019. The study comprised 23,805 individuals in the final analysis. Logistic regression and multivariable linear regression were applied to estimate the dietary pattern-specific associations between AM and hypertension or systolic/diastolic blood pressure (SBP/DBP). Restricted cubic spline regression was utilized to calculate the shape of the relationship between AM and the odds ratio of hypertension. After adjusting for multiple variables, women who had a history of AM > 14 years were associated with an increased risk of hypertension (OR 1.12, 95%CI 1.04–1.19) and elevated levels of SBP (*β* 0.90, 95%CI 0.41–1.38) compared with those with AM ≤ 14 years among the total population, and this association was still statistically significant when we further adjusted for body mass index (BMI). In participants with AM > 14 years, the odds ratio values of hypertension increased with increasing menarche age. After stratification by age at recruitment, the positive association between menarche age and hypertension only remained in the middle-aged group, and this association was not found in the young and old groups. After stratification by the Dietary Approaches to Stop Hypertension (DASH) score among the total population, the positive association between AM and hypertension was presented only in the low DASH score group; however, this association was not found in the high DASH score group. Women who have a history of AM > 14 years should pay close attention to blood pressure levels and incorporate the DASH diet more in order to achieve the early prevention of hypertension, especially middle-aged women.

## 1. Introduction

Hypertension is a major cause of cardiovascular diseases and mortality and has become an important public health issue worldwide [1,2]. Approximately 10.8 million deaths due to high systolic blood pressure (SBP) have been reported, making it a leading factor that accounted for deaths globally in 2019 [3]. In China, the prevalence of hypertension reached 23.2% in the Chinese adult population [4]. Therefore, it is necessary to identify potential risk factors for the prevention of hypertension.

Menarche is the first menstruation of a girl, representing the beginning of the reproductive life of a woman [5]. Findings from previous studies have suggested that age at menarche (AM) is associated with the risk of diabetes [6,7], obesity [8], and cardiovascular disease [9,10]. For the association between AM and hypertension, the results of previous epidemiologic studies have been inconsistent. Various studies have reported that AM is negatively associated with risk of hypertension [11,12]. Some researchers have demonstrated a positive association between AM and hypertension [13,14], while others have reported no such association [15]. These varying results may be due to genetic background [16], lifestyle behavior, and living environment among different populations. Meanwhile, limited attention has been paid to the association between AM and hypertension in southwest China.

A number of studies have conducted subgroup analyses according to region [12], age at recruitment [17], and body mass index (BMI) [14] to explore the different relationships that are possible between AM and hypertension among subgroups. However, only a few studies have examined stratified analyses by lifestyle behavior [18], and no studies have conducted subgroup analyses by dietary patterns when exploring the association between AM and hypertension. A dietary pattern, which is an important lifestyle behavior, may modify the association between AM and hypertension. Therefore, it is necessary to further explore whether there are different effects of AM on hypertension in different dietary pattern groups for the prevention of hypertension.

This study investigated the relationship between AM and hypertension and assessed this relationship by different dietary pattern groups in a large-scale Han Chinese population in southwest China.

## 2. Materials and Methods

### 2.1. Study Participants

We studied participants from the baseline survey of the China Multi-Ethnic Cohort Study, in the Chongqing and Chengdu region. We included 44,900 Han Chinese participants aged between 30 and 79 years from 17 districts in southwestern China from September 2018 to January 2019. Briefly, the study participants were recruited using a multistage, stratified cluster sampling method. Detailed data collection information has previously been presented [19]. Of these participants, the study included 24,422 women.

We excluded participants if they had incomplete information (*n* = 574), and those with AM earlier than 8 years or later than 22 years (*n* = 43). The analysis included 23,805 individuals in the evaluation of the association between AM and hypertension. Furthermore, we excluded participants who used antihypertensive drugs (*n* = 3137), and the final analysis included 20,668 individuals to evaluate the association between AM and SBP/DBP (Figure 1). All participants provided written informed consent prior to the survey. This survey was approved by the ethics committee of Sichuan University (No. K2016038). 

### 2.2. Data Collection and Laboratory Measurement

A face-to-face electronic questionnaire survey was conducted by interviewers to collect sociodemographic characteristics (age, marital status, education, socioeconomic status, etc.), lifestyle characteristics (smoking status, alcohol drinking, physical activity, dietary behavior, etc.), reproductive factors (AM, prior pregnancy status, use of oral contraceptive pills, etc.), and family history of disease. Participants were divided into three groups (<45 years, 45–65 years, >65 years). Marital status was grouped into married/cohabitated and other, including separated, divorced, widowed, and unmarried. Education level was classified into primary school or illiterate, junior high school, high school, junior college, and above. Socioeconomic status was determined by yearly household income (<CNY 12,000, CNY 12,000–19,999, CNY 20,000–59,999, CNY 60,000–99,999, ≥CNY 100,000). Smoking status was grouped into never, current, and ever smoker. Alcohol drinking status was categorized into never or hardly, occasionally, and often. Physical activity level was estimated based on metabolic equivalent (MET) with a dichotomized variable using 18.12 h/day as the cutoff point according to the median. A food frequency questionnaire was used to assess dietary behavior, which was calibrated through preliminary investigation. The Dietary Approaches to Stop Hypertension (DASH) score for each participant was based on food frequency information, which has been described in detail previously [20]. Additionally, the DASH diet has been proven to have beneficial effects on decreasing blood pressure (BP) [21]. In the present study, we assigned 7 kinds of food a score of 1 to 5 according to the quintile of the average food intake [22,23]. For whole grains, fresh fruits, fresh vegetables, legumes, and dairy products, a score of 5 was given for the highest quintile, and a score of 1 was given for the lowest quintile. For red meat products and sodium, this pattern of scoring was inverted. Then, all the food scores were summed to obtain an overall DASH score. Participants were categorized into a high score group (>22) and a low score group (≤22) based on the median of the DASH score.

After fasting for 12 h, venous blood samples were collected from all participants and tested at the Di’an Medical Laboratory Center. Dyslipidemia was diagnosed as a serum total cholesterol concentration of ≥6.2 mmol/L, triacylglycerol concentration of ≥2.3 mmol/L, low-density lipoprotein cholesterol concentration of ≥4.1 mmol/L, high-density lipoprotein cholesterol concentration of <1.0 mmol/L, or a self-reported diagnosis of dyslipidemia [24]. Participants were regarded as having diabetes if any of the following three items were met: fasting blood glucose level of ≥7.0 mmol/L, glycosylated hemoglobin percentage of ≥6.5%, or a self-reported diagnosis of diabetes [25]. Body mass index was calculated as weight in kilograms divided by the square of height in meters (kg/m^2^) and categorized into <24, 24–27.9 and ≥28 groups. According to the weight criteria for adults in China [26], overweight was defined as 24 ≤ BMI ≤ 27.9 kg/m^2^, and obesity was defined as BMI ≥ 28 kg/m^2^.

### 2.3. Age at Menarche Assessment

Information about self-reported age at menarche was collected through the question: “At what age did your first menstruation begin?” Further to this, all participants were divided into a ≤14 years group and a >14 years group based on the median as the cutoff point, with ≤14 years of age as the reference group.

### 2.4. Outcome Assessment

Each participant’s BP was measured by trained medical personnel using electronic sphygmomanometers. All the electronic sphygmomanometers were calibrated before measurement. Medical personnel strictly followed the American Heart Association’s standardized protocol for the measurement of BP [27]. All the BP measurements were performed three times with a seated, upright position after at least 5 min of rest. Systolic blood pressure and diastolic blood pressure (DBP) were calculated as the mean value of three measurements. In this analysis, hypertension was defined as having an average measured SBP ≥ 140 mmHg, DBP ≥ 90 mmHg, or self-reported diagnosis of hypertension [28,29]. 

### 2.5. Statistical Analysis

We carried out descriptive analyses of participants’ characteristics for the considered variables. All continuous variables were described as the median (Q1, Q3), while categorical variables were summarized as percentages. For the analysis of differences between the AM ≤ 14 years group and the AM > 14 years group, a Mann–Whitney U test was used to compare the differences among continuous variables, and a chi–square test was used to compare the differences among categorical variables. 

To evaluate the association between AM and hypertension, we used binary logistic regression to estimate the odds ratio (OR) and 95% confidence interval (CI) for hypertension. Multivariable linear regression was applied to investigate AM and its associations with SBP and DBP. In this analysis, we performed three models. The crude model was unadjusted for any covariates. In other words, the univariate analysis was used to evaluate the relationship between AM and blood pressure. Then, multiple regression analysis was carried out using models 1 and 2 in order to further understand the association between AM and blood pressure. Model 1 was adjusted for age, marital status, education level, yearly household income, smoking status, alcohol drinking status, physical activity, and DASH score (when stratification was carried out, DASH score was excluded), prior pregnancy status, use of oral contraceptive pills, family history of hypertension, dyslipidemia, and diabetes. Model 2 was also adjusted for BMI. As BMI may represent a mediation factor rather than a confounder associated with AM and hypertension [30,31], we adjusted BMI in a separate model. To examine potential effect modifiers, stratification analyses by the DASH score and age at recruitment were conducted in three models. 

All the analyses were carried out in SPSS 25.0, and a two-tailed *p* value < 0.05 was considered statistically significant. Restricted cubic spline regression was drawn using the ggplot2 and rms packages of the R software 4.1.1 to evaluate the shape of the relationship between menarche age and the odds ratio of hypertension by assigning knots at the 5th, 35th, 65th, and 95th percentiles; the reference value was AM of 14 years.

## 3. Results

### 3.1. General Characteristics of the Participants

The general characteristics of the study participants are shown in Table 1. Among the 23,805 participants, the mean menarche age was 14.36 years (range from 8 to 22 years), and the percentage of participants with AM later than 14 years was 40.18% (9565/23,805). Participants with AM > 14 years were statistically older than participants with AM ≤ 14 years, had a higher BMI, and were more likely to have experienced pregnancy, dyslipidemia, diabetes, and hypertension. In addition, compared to those with AM > 14 years, participants with AM ≤ 14 years were more likely to be married, had a higher level of education, had greater exposure to smoking, had a higher physical activity level and DASH score, were more likely to have a family history of hypertension and had higher oral contraceptive pill usage.

Overall, the prevalence of hypertension was 28.73% (6838/23,805) in our study population. Among the 6838 hypertension patients, the percentage of patients with self-reported diagnosis of hypertension was 51.36% (3512/6838), and the percentage of patients with hypertension detected by blood pressure tests in the survey was 48.64% (3326/6838).

### 3.2. Association between AM and Hypertension

Figure 2 displays the association between menarche age and hypertension among all study participants. In models 1 and 2, those with AM > 14 years had a higher odds ratio for hypertension (OR 1.12, 95%CI 1.04–1.19, *p* = 0.002; OR 1.15, 95%CI 1.07–1.24, *p* < 0.001, respectively) compared to participants with AM ≤ 14 years. Furthermore, after adjusting for multiple variables in model 1, each one-year higher menarche was associated with a 2% increase in hypertension (OR 1.02, 95%CI 1.01–1.04, *p* = 0.007); this association was still statistically significant when we further adjusted for BMI (OR 1.04, 95%CI 1.02–1.05, *p* < 0.001). 

The results for the cubic spline regression analyses are shown in Figure 3. The association between AM and hypertension was demonstrated as an approximately nonlinear tendency. Above 14 years, the odds ratio of hypertension steadily increased with the increasing menarche age versus the reference group (AM = 14 years) in models 1 and 2. However, below 14 years, the association between AM and hypertension was not statistically significant in models 1 and 2.

Figure 2 shows the association between AM and hypertension by the DASH score. We observed that the positive association between AM and hypertension was presented only in the low DASH score group. For the low DASH score group, those with AM > 14 years were more likely to have hypertension (OR 1.15, 95%CI 1.05–1.26, *p* = 0.003; OR 1.19, 95%CI 1.09–1.31, *p* < 0.001, respectively) compared to participants with AM ≤ 14 years in models 1 and 2. Furthermore, in models 1 and 2, each one-year higher menarche was associated with 3% and 4% increases in hypertension (OR 1.03, 95%CI 1.01–1.05, *p* = 0.013; OR 1.04, 95%CI 1.02–1.06, *p* < 0.001, respectively). However, this association was not found in the high DASH score group (*p* > 0.05).

The results of the subgroup analysis by age at recruitment concerning the association between AM and hypertension are shown in Table 2. In the three models, those with AM > 14 years remained at an increased risk of hypertension (OR 1.31, 95%CI 1.22–1.41, *p* < 0.001; OR 1.17 95%CI 1.08–1.27, *p* < 0.001; OR 1.22 95%CI 1.12–1.32, *p* < 0.001, respectively) in the middle-aged group compared to participants with AM ≤ 14 years; however, this association was not found in the young and old groups. Further stratification analysis by the DASH score among middle-aged participants showed that, after adjusting for multiple variables in models 1 and 2, those with AM > 14 years versus those with AM ≤ 14 years had a higher odds ratio for hypertension in the low DASH group than those in the high DASH group.

### 3.3. Association between AM and SBP/DBP

Multivariable linear regression analyses were applied to evaluate the association between AM and SBP/DBP; these findings are presented in Table 3. After adjusting for covariates in models 1 and 2, later menarche (AM > 14 years versus AM ≤ 14 years) was associated with 0.90 and 1.12 higher SBP (95%CI 0.41–1.38, *p* < 0.001; 95%CI 0.64–1.59, *p* < 0.001, respectively); however, this association was not significant for DBP (*p* > 0.05). Furthermore, each one-year increase in menarche was associated with 0.29 mmHg and 0.38 mmHg higher SBP (95%CI 0.17–0.41, *p* < 0.001; 95%CI 0.26–0.50, *p* < 0.001, respectively); however, similarly, there was no significant association between AM and DBP in models 1 and 2 (*p* > 0.05).

Stratification analyses by DASH score were conducted for AM and SBP/DBP; these results are also shown in Table 3. After adjusting for multiple variables in model 1, those with AM > 14 years versus those with AM ≤ 14 years had a higher beta (*β*) of SBP in the low DASH group (*β* 0.94, 95%CI 0.25–1.64, *p* = 0.008) than those in the high DASH group (*β* 0.83, 95%CI 0.16–1.50, *p* = 0.015); this association was still present when we further adjusted for BMI (*β* 1.21, 95%CI 0.53–1.90, *p* < 0.001; *β* 0.99, 95%CI 0.34–1.64, *p* = 0.003). Furthermore, each one-year higher menarche was associated with 0.39 mmHg (95%CI 0.22–0.56, *p* < 0.001) and 0.36 mmHg (95%CI 0.19–0.54, *p* < 0.001) higher SBP in the low and high DASH score groups, respectively, in model 2. However, the association between AM and DBP was no longer significant for the two DASH score groups in models 1 and 2 (*p* > 0.05).

## 4. Discussion

Our study found that later menarche (>14 years) was associated with an increased risk of hypertension and elevated levels of SBP after adjusting for multiple variables, including BMI, in southwest China among the total population. In addition, as AM increased, women had higher odds ratio values for hypertension in participants with AM > 14 years. To our knowledge, this is the first study to assess whether dietary patterns can modify the relationship between AM and hypertension among Han Chinese women aged 30 to 79 years. We found that the DASH diet modified the positive relationship between AM and hypertension, as this positive association was presented only in the low DASH score group among all study participants. However, this association was not found in the high DASH score group. The results provide novel individual-level prevention measures for hypertension in participants with AM > 14 years.

Our findings were consistent with previous studies that found that later menarche age was associated with a higher risk of hypertension [13,14,17]. The cubic spline regression results further identified a positive association between AM and hypertension, as the odds ratio for hypertension steadily increased with the increasing menarche age in AM > 14 years participants. However, no significant association was found for AM and hypertension in participants with AM ≤ 14 years. This is the reason why all the participants in the present study were divided into two groups based on AM 14 years as the cutoff point. These potential mechanisms can partly explain the positive association between AM and hypertension. Studies have shown that late menarche is associated with low levels of estrogen, ovarian hormone, and growth hormone [32,33]; these hormones can reduce blood pressure and protect women from hypertension and cardiovascular disease [34,35,36,37]. However, our findings were inconsistent with a few previous studies. An analysis of 7893 urban women aged 45 years and older found that AM was not associated with hypertension [12]. Zhang et al. enrolled 15,361 rural Chinese postmenopausal women from Henan, China and found early menarche was associated with a higher risk of hypertension [31]. These inconsistent findings might be related to differences in sample size, region, menarche age grouping, and considered confounders; for example, dietary factors are strongly associated with hypertension. However, few previous studies have adjusted for diet status, perhaps leading to statistical bias. To our knowledge, the present study is the first study to examine the relationship between AM and hypertension after adjusting for dietary pattern. 

Compared with previous studies, we innovatively conducted stratification analysis by DASH score. The results revealed that those who had a history of AM > 14 years with a low DASH score remained at an increased risk of hypertension compared with AM ≤ 14 years women; however, this positive association was not found in the high DASH score group among all study participants. The difference in results of the two groups indicated the necessity to stratify by DASH score and implied that dietary patterns play an important role in the relationship between AM and hypertension. In other words, the DASH diet may have modified the relationship between AM and hypertension. To our knowledge, no study has explored the possible mechanisms of a DASH diet effect for AM and hypertension. The DASH diet is rich in legumes, which are the only dietary source of isoflavones, imparting estrogen’s physiological effects into humans [38]. Therefore, the DASH diet’s inclusion of a high amount of legumes may mediate the effect that late menarche has on increased risk of hypertension. This may help to partially explain the absence of any positive association for women with late menarche in the high DASH score group in the present study. Of course, other foods included in the DASH diet may also be related to the association between AM and hypertension, such as fresh fruits, fresh vegetables, and dairy products. Therefore, more evidence on the mechanisms of the different relationships between AM and hypertension in the high DASH score and low DASH score groups is needed in the future. As mentioned above, the results suggested that late menarche women should pay more attention to blood pressure, and those who have a history of AM > 14 years should incorporate more DASH diet foods for prevention of hypertension. This may provide novel individual-level prevention measures for hypertension in participants with AM > 14 years.

Moreover, we conducted stratified analysis by age at recruitment. The result revealed that those with AM > 14 years remained at an increased risk of hypertension compared with participants with AM ≤ 14 years in the middle-aged group; however, this positive association was not found in the young and old groups. The difference in the results of the three groups indicated the necessity to stratify by age at recruitment. Additionally, this interesting result implies that age at recruitment plays an important role in the relationship between AM and hypertension. There are two possible reasons why the positive association was not present in young or old people. The first reason may be the different metabolic function in young and old people compared with the middle-aged population. It may also be related to the data structure itself, where the prevalence of hypertension is very low (6.85%, 538/7858) among young people, and the proportion of those with AM ≤ 14 years and AM > 14 years is not balanced (25.44%, 74.56%) among old people, which may make it difficult to identify the relationship between age at menarche and hypertension. Furthermore, a stratified analysis by the DASH score was conducted in the middle-aged population, and we found that those with AM > 14 years versus those with AM ≤ 14 years had a higher odds ratio for hypertension in the low DASH group than those in the high DASH group among middle-aged participants. This result also partly supports the speculation that the DASH diet may have a potential modification effect for the positive relationship between menarche age and hypertension.

In addition, we evaluated the associations between AM and SBP/DBP in the entire population. Our results showed that later menarche was associated with higher SBP, but this association was not significant for DBP. The reasons for the different effects of AM on SBP and DBP were unknown, and this needs to be explored further. In the present study, we also examined whether dietary patterns modified the associations between AM and SBP/DBP. Our findings showed that the strength of association for AM and SBP in the low DASH score group was stronger compared with the high DASH score group, which implied that the DASH diet pattern may decrease the effect of AM on SBP. Those results further support the theory that the DASH diet may mediate the effect that late menarche has on increased risk of hypertension, possibly related to food included in the DASH diet, such as legumes. 

One of the strengths of the present study is that it included an exploration of the relationship between AM and hypertension for the first time using large-scale cross-sectional study data in southwest China. Chongqing and Chengdu are adjacent cities, located in the southwest of China, with similar lifestyle habits, environmental conditions, and economic levels; thus, large samples from Chongqing and Chengdu are representative of southwest China. Moreover, we comprehensively considered related confounders to explore the relationship between AM and hypertension in this study, particularly dietary patterns, which have not been taken into account in previous studies. In addition, taking the influence of dietary patterns on the association between AM and hypertension into consideration, we innovatively conducted this analysis, stratifying by the DASH score, and found that dietary patterns may modify the relationship between AM and hypertension. However, this study has several limitations. Firstly, our study data were cross-sectional results from a cohort study; cross-sectional data analysis limited our ability to explore the causal relationship between AM and hypertension, for which prospective studies will be needed. Secondly, AM was retrospectively assessed by a question, which may cause recall bias. However, previous studies have found that AM by recall highly matched original adolescent reports [39,40], suggesting that AM by recall could be applied in scientific research. In addition, AM by recall has also been used in other similar studies of the relationship between AM and hypertension [14,30]. Thirdly, the present study found that SBP increased by 0.29 mmHg with each one-year increase in menarche, which is clinically insignificant. Thus, caution should be taken in clinical applications. Fourthly, our study was limited due to age and ethnicity, so extrapolating the results to other age groups and ethnicity populations should be noted.

## 5. Conclusions

In the entire population, later menarche (>14 years) was associated with an increased risk of hypertension and elevated levels of SBP compared with early menarche (≤14 years), and the odds ratio values of hypertension increased with increasing menarche age in participants with AM > 14 years. After stratification by age at recruitment, the positive association between menarche age and hypertension only remained in the middle-aged group. In addition, the DASH diet may modify the relationship between AM and hypertension, as this positive association was present only in the low DASH score group after stratification by the DASH score among all study participants; however, this association was not found in the high DASH score group. Our results indicate that women who have a history of AM > 14 years should pay close attention to their blood pressure levels and should incorporate more DASH diet foods in order to achieve early prevention of hypertension, especially middle-aged women.

## Figures and Tables

**Figure 1 nutrients-14-01610-f001:**
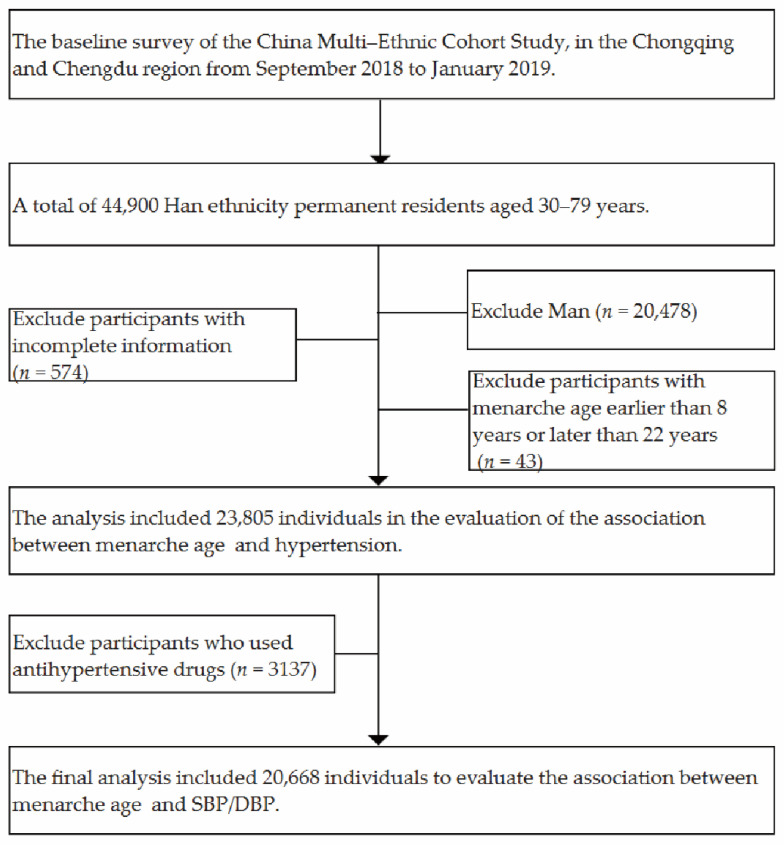
Data cleaning flowchart.

**Figure 2 nutrients-14-01610-f002:**
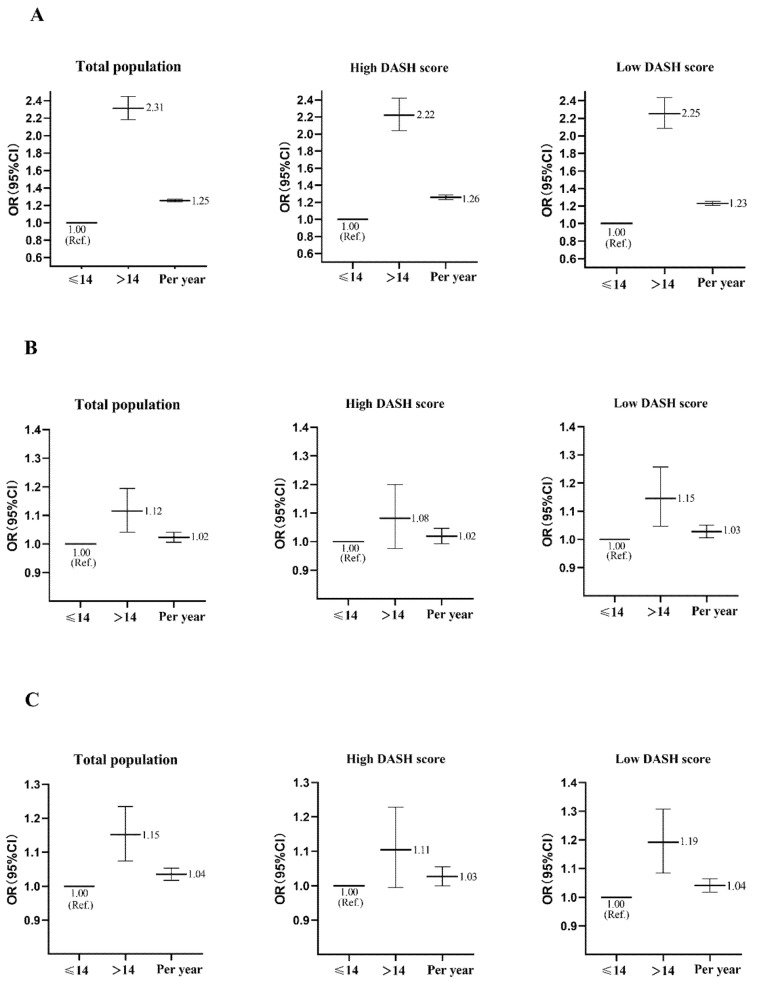
Odds ratio (OR) and 95% CI for hypertension according to binary categories (AM > 14 years vs. AM ≤ 14 years) or per year increase in AM among individuals in the total population and high and low DASH score groups. (**A**) Results using the crude model regression analyses. (**B**) Results using model 1 regression analyses. (**C**) Results using model 2 regression analyses. Ref. indicates the reference group.

**Figure 3 nutrients-14-01610-f003:**
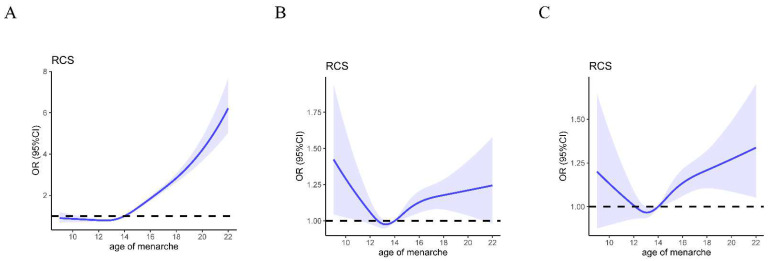
Curve of the association between hypertension risk and per year increase in menarche age. The solid line represents the estimated odds ratio for hypertension risk and the shaded areas represent 95%CI. (**A**) Results using the crude model cubic spline regression analyses. (**B**) Results using the model 1 cubic spline regression analyses. (**C**) Results using the model 2 cubic spline regression analyses.

**Table 1 nutrients-14-01610-t001:** General characteristics of the participants according to menarche age.

Characteristics	Total (*n* = 23,805)	Age at Menarche		*p*-Value
		≤14 (*n* = 14,240)	>14 (*n* = 9565)	
Age (years)				<0.001
<45	33.01	45.29	14.73	
45–65	53.76	49.09	60.72	
>65	13.23	5.63	24.55	
Married or living with partner	87.02	89.02	84.04	<0.001
Education level				<0.001
Primary school or illiterate	35.20	23.56	52.52	
Junior high school	31.47	32.73	29.59	
High school	17.22	21.04	11.54	
Junior college and above	16.11	22.67	6.35	
Yearly household income (CNY)				<0.001
<12,000	10.07	7.39	14.05	
12,000–19,999	13.29	11.66	15.71	
20,000–59,999	36.96	35.48	39.17	
60,000–99,999	20.82	22.50	18.33	
≥100,000	18.86	22.97	12.73	
Smoking				<0.001
Never	97.72	97.23	98.45	
Current	1.91	2.36	1.23	
Ever	0.37	0.41	0.31	
Alcohol drinking				<0.001
Never or hardly	62.83	58.79	68.84	
Occasionally	33.95	38.48	27.20	
Often	3.22	2.72	3.95	
Physical activity (MET h/day)				<0.001
≤18.12	50.01	47.83	53.26	
>18.12	49.99	52.17	46.74	
DASH score				<0.001
≤22	50.30	45.71	57.12	
>22	49.70	54.29	42.88	
Family history of hypertension	40.66	42.79	37.48	<0.001
Use of oral contraceptive pills	21.55	26.80	13.74	<0.001
Pregnancy	98.70	98.29	99.31	<0.001
BMI (kg/m^2^)				<0.001
<24	51.68	55.01	46.72	
24–27.9	35.78	33.67	38.92	
≥28	12.54	11.33	14.35	
Hypertension	28.73	21.74	39.12	<0.001
SBP (mmHg)	122.00 (110.67, 137.33)	118.67 (108.67, 131.67)	128.67 (115.33, 144.00)	<0.001
DBP (mmHg)	75.33 (69.00, 82.67)	74.67 (68.33, 81.67)	76.67 (70.00, 84.00)	<0.001
Dyslipidemia	26.93	23.69	31.76	<0.001
Diabetes	10.09	7.63	13.77	<0.001

Note: continuous data were described as the median (Q1, Q3), and statistical significance was assessed by the Mann–Whitney U test. Categorical data were summarized as percentages (%), and statistical significance was assessed by a chi-square test.

**Table 2 nutrients-14-01610-t002:** Associations between menarche age and hypertension by age at recruitment.

Variables	Crude Model		Model 1		Model 2	
	OR (95%CI)	*p*-Value	OR (95%CI)	*p*-Value	OR (95%CI)	*p*-Value
<45						
Total population	0.97 (0.77, 1.22)	0.774	0.85 (0.67, 1.07)	0.167	0.88 (0.69, 1.12)	0.292
High DASH score	0.89 (0.63, 1.27)	0.529	0.82 (0.57, 1.17)	0.274	0.89 (0.61, 1.28)	0.526
Low DASH score	1.00 (0.73, 1.35)	0.980	0.87 (0.64, 1.20)	0.408	0.87 (0.63, 1.21)	0.406
45–65						
Total population	1.31 (1.22, 1.41)	<0.001	1.17 (1.08, 1.27)	<0.001	1.22 (1.12, 1.32)	<0.001
High DASH score	1.24 (1.11, 1.39)	<0.001	1.15 (1.02, 1.30)	0.023	1.18 (1.05, 1.34)	0.007
Low DASH score	1.29 (1.16, 1.42)	<0.001	1.19 (1.07, 1.32)	0.001	1.24 (1.12, 1.38)	<0.001
>65						
Total population	0.97 (0.82, 1.15)	0.717	1.04 (0.87, 1.24)	0.687	1.03 (0.86, 1.24)	0.717
High DASH score	0.90 (0.70, 1.14)	0.368	0.92 (0.71, 1.19)	0.530	0.90 (0.69, 1.17)	0.432
Low DASH score	1.05 (0.83, 1.34)	0.683	1.13 (0.88, 1.46)	0.338	1.14 (0.89, 1.48)	0.305

**Table 3 nutrients-14-01610-t003:** Associations between menarche age and SBP/DBP.

Variables	Crude Model		Model 1		Model 2	
	*β* (95%CI)	*p* Value	*β* (95%CI)	*p* Value	*β* (95%CI)	*p* Value
SBP ^a^						
Total population	7.38 (6.89, 7.87)	<0.001	0.90 (0.41, 1.38)	<0.001	1.12 (0.64, 1.59)	<0.001
High DASH score	6.46 (5.78, 7.15)	<0.001	0.83 (0.16, 1.50)	0.015	0.99 (0.34, 1.64)	0.003
Low DASH score	7.41 (6.71, 8.11)	<0.001	0.94 (0.25, 1.64)	0.008	1.21 (0.53, 1.90)	0.001
DBP ^a^						
Total population	1.48 (1.21, 1.76)	<0.001	−0.05 (−0.35, 0.25)	0.744	0.08 (−0.22, 0.37)	0.613
High DASH score	1.11 (0.71, 1.51)	<0.001	−0.29 (−0.71, 0.13)	0.178	−0.20 (−0.61, 0.21)	0.342
Low DASH score	1.56 (1.16, 1.95)	<0.001	0.19 (−0.23, 0.61)	0.384	0.35 (−0.07, 0.76)	0.103
SBP ^b^						
Total population	2.08 (1.96, 2.20)	<0.001	0.29 (0.17, 0.41)	<0.001	0.38 (0.26, 0.50)	<0.001
High DASH score	1.94 (1.77, 2.12)	<0.001	0.30 (0.12, 0.47)	0.001	0.36 (0.19, 0.54)	<0.001
Low DASH score	2.00 (1.84, 2.17)	<0.001	0.28 (0.11, 0.45)	0.001	0.39 (0.22, 0.56)	<0.001
DBP ^b^						
Total population	0.37 (0.30, 0.44)	<0.001	−0.06 (−0.13, 0.02)	0.149	−0.01 (−0.08, 0.07)	0.896
High DASH score	0.32 (0.21, 0.42)	<0.001	−0.10 (−0.21, 0.01)	0.085	−0.06 (−0.17, 0.05)	0.285
Low DASH score	0.35 (0.26, 0.45)	<0.001	−0.02 (−0.12, 0.09)	0.729	0.04 (−0.06, 0.15)	0.413

Note: ^a^ age at menarche as categorical variable; ^b^ age at menarche as continuous variable.

## Data Availability

Our study relied on data from the China Multi-Ethnic Cohort Study. The summary dataset used and or analyzed during the current study are available from the corresponding author upon reasonable request.

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
