# Peer review of "Associations between Age at Menarche and Dietary Patterns with Blood Pressure in Southwestern Chinese Adults"

_nutrients, 2022, doi:10.3390/nu14081610_

Round 1
Reviewer 1 Report
Dear Editor,
I carefully read the manuscript by Chen et al. that is of potential interest for the readers of the Journal.
My comments and suggestions for the authors are the following:
- Page 2, Line 68: A flow-chart should be included here for more clarity.
- Page 3, Line 122: "Hypertension blood pressure" is an incorrect expression. The authors should refer to "hypertension" only.
- Page 3, Line 131: Statistical analysis is roughly described. For example, was the "crude model" an univariate analysis?
- Table 1: The authors should report separately information regarding the incidence of hypercholesterolemia and hypertriglyceridemia. The definition of "dyslipidemia" little suits the context.
- English language needs to be carefully revised and improved.
- The authors should consider to refer to doi: 10.1007/s40292-021-00474-6 and doi: 10.1016/j.numecd.2020.03.005 in their manuscript.
- The limitations of the study should be more deeply discussed in the manuscript.
Author Response
Sincerest thanks for the reviewer for the positive and suggestive comments on our manuscript. Those comments are all valuable and very helpful for improving our paper. We have carefully revised the paper based on those comments and recommendations. The detailed point-by-point responses to the comments were listed in attach file.

Reviewer 2 Report
“Associations of age at menarche and dietary pattern with blood pressure in southwestern Chinese adults”
Overall score: 60/100; major revision
The presented article is a cross-sectional study involving a homogeneous population of southwestern China. The authors of the publication did significant work to statistically evaluate the relationship between the age of onset of first menstruation and hypertension. Of note is the researchers' diligence in evaluating the many factors that interfere with the possible relationship and examining the possible influence of diet. The article is understandable but needs intensive language improvement. Below are my comments.
Major Comments:
- In the presented cohort of patients, there is a very strong relationship between the age of the respondent and the age of first menstruation. Nearly half of the respondents in the group of menarche <14 years , were in the group of age <45 years. The authors partially accounted for this fact by using further multivariate analyses, but I believe that such a large disproportion of young women in the menarche group <14 years may strongly interfere with further analyses. In addition, younger women, may differ from older women by other factors that are not included in Table 1. Therefore, I suggest performing separate analyses in stratified groups of women - e.g., age quartiles, to see if the main results of the paper will hold. Such a sensitivity analysis could further strengthen the final results of the paper.
- The menarche age of 14 was chosen arbitrarily. The authors based on cubic spline regression (Figure 2, first graph on the left) argue that this is the correct cut-off point in that the relationship becomes significant from age 14. However, the next two graphs (Figure 2, middle and right panels), which incorporated confounding factors, show the lowest point around age 13. Shouldn't this point be chosen as the menarche cutoff for the entire analysis instead of age 14?
- Did the study collect information on the use of hypertension medication - was this taken into account when measuring sBP and dBP during the survey? Women treated for hypertension should be excluded from the analysis of sBP and dBP in relation to menarche (Table 2).
- The relationship between sBP and age of menarche is clinically insignificant - 0.38mmHg for each year of menarche - please comment on this and possibly include this in the study limitations
Minor Comments:
- Please provide the mean value of sBP and dBP in Table 1.
- There is a lack of information on how many cases of HBP were detected during BP testing associated with the survey, and how many cases were HBP given by the survey respondent.
- I ask the authors to comment on the low physical activity in the survey group - 18.12 MET x min/week seems absurdly low. Is this comparable to previous studies and how might this affect the results?
Author Response

(The authors gave the same response as above.)

Round 2
Reviewer 1 Report
Dear Editor,
I carefully read the revised version of the manuscript that is significantly improved in comparison with the original version.
Reviewer 2 Report
The authors have addressed all of the comments and I believe that the article should be published in "Nutrients"
I have two minor comments regarding spelling errors:
- Line 822 and 823:
"and the proportion of those with AM ≤ 14 years and AM ≤ 14 years is not balanced (25.44%, 74.56%)"
should be
"with AM ≤ 14 years and AM > 14 years"
2: The unit of physical activity should be MET-hour/week instead of MET-hour/day. Please check this and replace throughout the text if appropriate.